# Pyrazinacenes exhibit on-surface oxidation-state-dependent conformational and self-assembly behaviours

David Miklík[1,2,12], S. Fatemeh Mousavi[3,12], Zuzana Burešová[2], Anna Middleton[4], Yoshitaka Matsushita [5], Jan Labuta [1], Aisha Ahsan[3], Luiza Buimaga-Iarinca[6], Paul A. Karr[7], Filip Bureš[2], Gary J. Richards [1,8✉], Pavel Švec[1], Toshiyuki Mori [9], Katsuhiko Ariga[1,10], Yutaka Wakayama[1], Cristian Morari[6], Francis D'Souza [4✉], Thomas A. Jung [11✉] & Jonathan P. Hill [1✉]

Acenes and azaacenes lie at the core of molecular materials' applications due to their important optical and electronic features. A critical aspect is provided by their heteroatom multiplicity, which can strongly affect their properties. Here we report pyrazinacenes containing the dihydro-decaazapentacene and dihydro-octaazatetracene chromophores and compare their properties/functions as a model case at an oxidizing metal substrate. We find a distinguished, oxidation-state-dependent conformational adaptation and self-assembly behaviour and discuss the analogies and differences of planar benzo-substituted decaazapentacene and octaazatetracene forms. Our broad experimental and theoretical study reveals that decaazapentacene is stable against oxidation but unstable against reduction, which is in contrast to pentacene, its C–H only analogue. Decaazapentacenes studied here combine a planar molecular backbone with conformationally flexible substituents. They provide a rich model case to understand the properties of a redox-switchable π-electronic system in solution and at interfaces. Pyrazinacenes represent an unusual class of redox-active chromophores.

[1] International Center for Materials Nanoarchitectonics, National Institute for Materials Science, Tsukuba, Ibaraki, Japan. [2] Institute of Organic Chemistry and Technology, Faculty of Chemical Technology, University of Pardubice, Pardubice, Czech Republic. [3] Department of Physics, University of Basel, Basel, Switzerland. [4] Department of Chemistry, University of North Texas, Denton, TX, USA. [5] Research Network and Facility Services Division, National Institute for Materials Science, Tsukuba, Ibaraki, Japan. [6] National Institute for Research and Development of Isotopic and Molecular Technologies (NIRDIMT), Cluj-Napoca, Romania. [7] Department of Physical Sciences and Mathematics, Wayne State College, Wayne, NE, USA. [8] Department of Applied Chemistry, Graduate School of Engineering and Science, Shibaura Institute of Technology, Saitama-shi, Saitama, Japan. [9] Center for Green Research on Energy and Environmental Materials, National Institute for Materials Science, Tsukuba, Ibaraki, Japan. [10] Graduate School of Frontier Sciences, The University of Tokyo, Kashiwa, Japan. [11] Laboratory for Micro- and Nanotechnology, Paul Scherrer Institute, Villigen, Switzerland. [12]These authors contributed equally: David Miklík, S. Fatemeh Mousavi. ✉email: richards@shibaura-it.ac.jp; francis.dsouza@unt.edu; thomas.jung@psi.ch; jonathan.hill@nims.go.jp

Redox chemistry, i.e. the controlled oxidation/reduction of atoms, also within molecules affecting their chemical, electronic and optical properties, has been a significant driver e.g. for synthetic and materials chemistry[1–3] and for the ability to modulate oxidation states and organic/inorganic dye structures among many other properties. An increasingly important challenge is to control site-specific redox functionality and physicochemical properties of molecules within supramolecular architectures[4,5], especially at surfaces and interfaces[6–10] where access to such functionality and the corresponding applications would be facilitated even at the molecule level[11,12]. Biochemical systems have evolved towards exceptional levels of control over the supramolecular and metal coordination mechanisms governing chemical reactions[13]. These structures, however, are usually too delicate for technological applications. On the other hand, the engineering of organic interfaces and electrochemically or optically functional surfaces also requires precise control over the assembly of redox functional systems on surfaces[14]. Thus, established structure–function relationships of molecular materials known from solution need to be re-evaluated to predict and understand the interface-specific chemical, electronic, optical and mechanical properties of any newly synthesized molecules. Along these lines, we here discuss a special class of heteroacenes in-solution and on surfaces, in particular for their complex and interesting redox activity and supramolecular chemistry by taking a broad experimental approach in conjunction with ab initio property prediction techniques.

We report N-heteroacenes[15–18], the pyrazinacenes[19–21], composed of linearly fused pyrazine rings, possessing multi-stability of redox/photo-redox chemistry in-solution and when adsorbed at a Cu(111) surface. The physicochemical properties of these molecules are discussed in the context of their modified redox state before and after on-surface reactions: we link their different observed oxidation states to their conformation, their configurations and their aggregation on surfaces. The complexity of this system derives from its molecular architecture, which combines a planar molecular backbone with substituents exhibiting conformational freedom and a redox-variable π-electronic system. The first step of oxidation enables cooperative conformational adaptability and aggregation. The second step leads to chemisorption prohibiting self-assembly of the molecules due to strong interactions with the substrate. Indeed, it is remarkable that a redox-active molecular module can be made that does not polymerize by reaction or coordination on surfaces but rather expresses the same conformational states in-solution and on surfaces also involving assembly into 2D islands. The non-planar molecular morphology together with the three-state redox system are key to the complexity observed and have been confirmed independently using high-resolution scanning tunneling microscopy (STM) and ab initio density functional theory (DFT) simulations (see Supplementary Information (SI) for experimental and computational details).

The pyrazinacenes used here are stable, length-tunable redox-active molecular modules, which can be activated in three steps. Remarkably, pyrazinacenes are sufficiently stable at each stage of oxidation in different media and in different states of aggregation[19–21]. The native and oxidized forms are chemically highly stable and do not undergo auto-catalysed or surface-catalysed reactions. Similar systems can be built/designed, in particular if larger 'protective' substituents can be incorporated into less inert multi-stage redox systems as seen in protein biochemistry where the bulky polypeptide backbone protects redox centres from prevailing solution conditions and defines its selectivity[22]. Delocalized electron systems comprising N and C π-electronic states[23,24] qualify the pyrazinacenes for the investigation of their light absorption as a chromophore and their redox

behaviour, also within specific supramolecular and on-surface architectures.

## Results

**Synthesis**. Pyrazinacene-type molecules bearing N-substituents have been prepared previously and are known to be highly fluorescent molecules with potential applications beyond the usual acene-type applications[19–21,25–29]. Decaazapentacenes (DAP, see Fig. 1a) had not been previously prepared in a form with accessible aromatic or other planar oxidized states. For the present study, a synthesis (Fig. 1b) was developed leading initially to dihydro derivatives, which were expected to be susceptible to oxidation in situ at a Cu(111) surface[30]. Synthesis was undertaken using a common precursor, dichloro-3 (see Supplementary Information). Initial syntheses of H$_2$DAP 1 was by condensation of dinitrilo-3 with diaminomaleonitrile followed by oxidation of the resulting 5,10-dihydrohexaazaanthracene with manganese dioxide. Hexaazaanthracene 5 was then condensed with 2,3-diaminopyrazine 6 yielding H$_2$DAP 1. Improved synthetic yields, however, could be obtained by converting dichloro-3 into diamine 7 followed by condensation of dichloro-3 with 7. Octaazatetracene 2 is available by condensation of dichloro-3 with 2,3-diaminopyrazine 6. Crystals of 2 could be obtained by diffusion of hexane into a solution of 2 in tetrahydrofuran and the X-ray crystal structure (Supplementary Fig. 1) reveals that the two protons of 2 are 'delocalized' over the four central nitrogen atoms with a molecule of THF hydrogen bonded with disordering corresponding to the NH delocalization. A DFT calculated structure of 2 (Supplementary Fig. 2) is remarkable in its similarity notwithstanding the delocalized NH protons) and the calculated structure of 1 is also shown in Supplementary Fig. 3. Both 1 and 2 can be deprotonated in a stepwise manner (Supplementary Figs. 4 and 5) leading to persistent highly fluorescent solutions. 1 and 2 can also be oxidized to 1-ox and 2-ox, respectively, by treatment with PbO$_2$ in dichloromethane or by heating in the presence of Cu(111) (vide infra).

**On-surface reactivity**. The initial impetus for this work was to study pyrazinacenes at metal surfaces in particular to observe the effects of the dihydropyrazine unit, which is known to be formally electronically delocalized yielding a planar molecular morphology[30]. 1 and 2, due to their similar structures, appear also similar when studied by STM on a Cu(111) surface, which we have used as a model substrate to investigate their supramolecular self-assembly and reactivity at the interface, as it evolves with progressive chemical conversion. Details of 2 are shown (Fig. 2; for compound 1 see Supplementary Fig. 6; see also Supplementary Figs. 7 and 8). For 2, the initially deposited molecules exhibit a double-lobed morphology (Fig. 2a, d). The almost exclusive presence of individual adsorbates indicates repulsive electrostatic interactions occurring after polarization or charge transfer by the interaction between the adsorbate and the substrate. We attribute these non-aggregating forms of 2 (and 1) to their native oxidation state containing one reduced pyrazine ring.

The molecules are adsorbed in registry with the substrate with 6 different orientations (Supplementary Fig. 9a), as expected for a chiral conformer (×2) on a threefold (×3) symmetric substrate surface. Note that the chiral conformers are caused by the relative twisting of the phenyl groups with each denoted as 'S' and 'mirror-S' in Supplementary Fig. 9b. As deposited, molecules exhibit an in situ XPS N1s spectrum (Fig. 2g) containing two peaks (NH (399.7 eV) and pyridine N (398.3 eV)) with intensities in agreement with the stoichiometry of nitrogen atoms in 2. These XPS energy values are highly consistent with those previously observed for similar enamine NH and pyridinic

**Fig. 1 Chemical structures and synthesis of pyrazinacenes. a** General formula for pyrazinacenes and structure of decaazapentacene. **b** Synthesis of **1** and **2** used in this work. R = $C_6H_5$ (**1**, **2**), 4-tBuC$_6$H$_4$ (**1**-tBu); X = nitrile or chloride. Reaction conditions: (i) DMSO, Na$_2$CO$_3$, 90 °C, 4 h, 47% (ii) CCl$_4$:MeCN:H$_2$O (2:2:1), RuO$_2$, NaIO$_4$, r.t., 1 h, 50–60%. (iii) DMSO, Na$_2$CO$_3$, 120 °C, 4 h, 10 %. (iv) DMSO, K$_2$CO$_3$, 120 °C, 4 h, 35 %. (v) PbO$_2$/CH$_2$Cl$_2$ or Cu(111)/150 °C.

N atoms whose signals appear, respectively, at 399.6[31,32] and 398.3 eV[32].

To our surprise, **1** or **2** gradually assemble into linear arrays after annealing at 150 °C (Fig. 2b, e), which are formed by molecules arranged with their long axes aligned parallel with an offset. Note that small amounts of such linear arrays are already visible after preparing and holding the sample at room temperature; see Supplementary Fig. 10. These linear arrays are stabilized by C–H…N hydrogen bonds involving phenyl rings and pyrazine rings as they are known from crystal structures of similar molecules[20]—see also Supplementary Figs. 11 and 12. The long symmetry axes of the molecular building blocks are canted at an angle of 75° with respect to the axis of the line structures. The phenyl groups of **1** and **2** can be accommodated in different conformations within the self-assembled line structure (Supplementary Figs. 13, 14 and Supplementary Note 1). XPS data (Fig. 2h and Supplementary Fig. 6h) reveal that a single type of pyridine-type N atom (N1s 398.3 eV[32]) is present in molecules contained in the line structure indicating that dehydrogenation of the pyrazinacene core of **2** has occurred yielding what we assign as **2**-ox. Annealing at 150 °C appears to increase the rate of line formation but also leads to the appearance of unusual double-lobed structures different from the starting morphology. To investigate this, we annealed at higher temperature (300 °C, see Fig. 2c), which resulted in the elimination of the line structures. In contrast, molecules were distributed across the surface, similar to the as-deposited state, albeit they exhibit exclusively the new double-lobed profile shown in Fig. 2f and Supplementary Fig. 6f. We assign the double-lobe/two dark satellites transition to molecules that have been further oxidized, this time

cyclodehydrogenated, with concurrent release of hydrogen into the vacuum. Cu(111) is known to be capable of acting as an oxidizing agent and in solutions, **2** can be easily oxidized to **2**-ox even by mild oxidizing agents. Cu(111) is also known to act as a cyclodehydrogenating agent at elevated temperatures[33] such that **2**-ox is converted to **2**-ox$_2$ containing terminal phenanthrene units and re-modifying the charge transfer and on-surface self-assembly. The STM profiles of **1** and **2** (Figs. 2 and 3 and Supplementary Figs. 7 and 8) appear considerably 'flattened' after extended annealing at higher temperatures and contrast strongly with those of the as-deposited molecules. During this annealing-induced transition, the phenyl substituents take a symmetric, co-planar conformation. Correspondingly, the conformational enantiomers are no longer visible. We attribute this to chemical conversion on the Cu substrate involving cyclodehydrogenation of the peripheral 1,2-diphenylbenzene units of **2** activated at 300 °C by the presence of the Cu(111) surface. Notably this reaction yields tetrabenzo[a,c,n,p]-5,7,9,11,18,20,22,24-octaaza-hexacene whose dimensions and profile fit well with those observed. There is also no substantial variation in the XPS spectra (pyridine-type N atom, N1s 398.3 eV[32]; Fig. 2i and Supplementary Fig. 6i) for the final annealed states **1**-ox$_2$ and **2**-ox$_2$ confirming that the respective decaazapentacene and octaazate-tracenene cores remain intact with all nitrogen atoms at energy consistent with their pyridinic form.

As summarized in Fig. 3, the pyrazinacene molecules **2** (**1** behaves essentially analogously as shown in Supplementary Figs. S13 and S14), undergo two oxidative transformations on Cu(111). The first oxidation results in **2**-ox (**1**-ox) (whose non-co-planar phenyl substituents remain visible), which assembles to

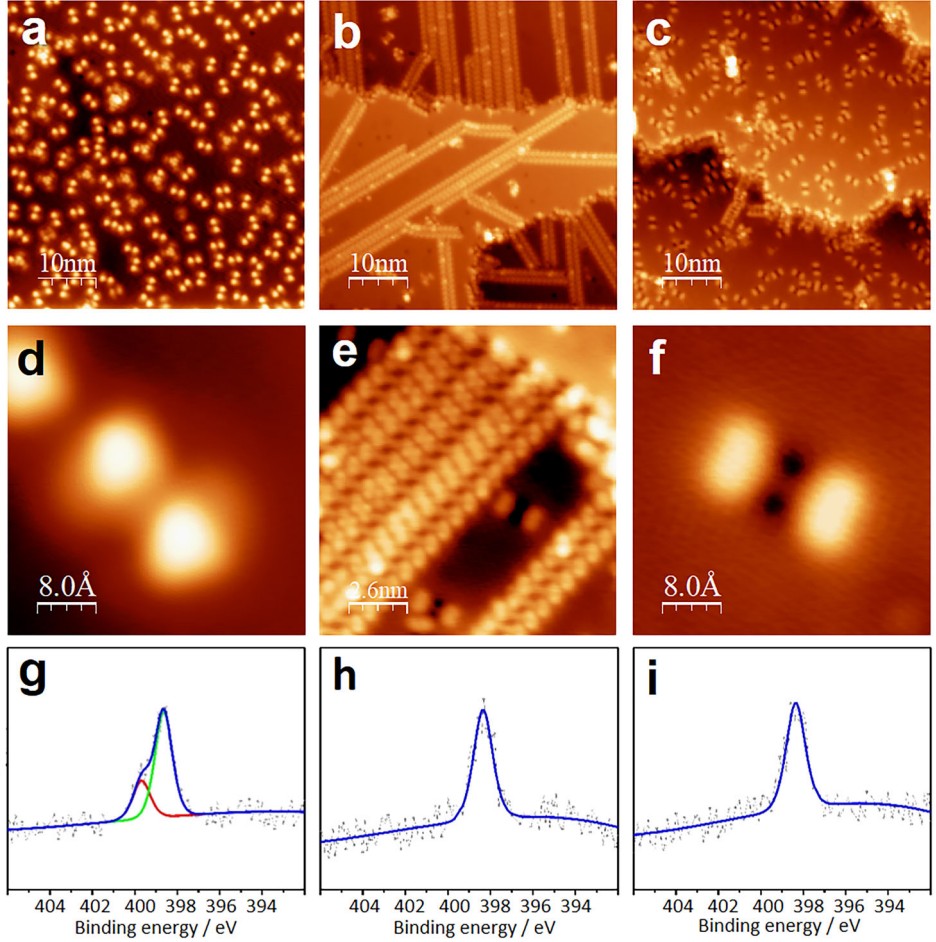

**Fig. 2 Scanning tunneling micrographs showing the evolution of conformation and assembly of pyrazinacene 2 with its progressive chemical conversion/oxidation on Cu(111) of 2 on Cu(111) substrate. a** Double-lobed shape of randomly distributed, individually adsorbed molecules of **2** observed after deposition onto the substrate held at room temperature (Image taken at 5 K: 50 × 50 nm, 10 pA, 1 V). **b** By thermal activation at ambient temperature, linear arrays form gradually within a few hours indicating a complex multi-step conversion process. This process is accelerated at elevated temperatures up to 150 °C. (Image: 50 × 50 nm, 30 pA, 500 mV). **c** Modified individual adsorbates observed after annealing at temperatures between 150 °C and 300 °C (Image: 50 × 50 nm, 10 pA, 1 V). The dark zone at the position of the molecular backbone and the reduced apparent height of the molecules (see also Fig. 3 and Supplementary Figs. 7 & 8) indicate their strengthened interaction with the substrate after annealing. **d–f** Magnified images of species shown in **a–c**, respectively. **d** The double-lobe shaped monomers are chiral as marked by an 'S' (Image: 4 × 4 nm, 50 pA, 1 V). **e** Line structures showing packed monomers (Note that the spots are identified as phenyl rings and the lines correspond to the acene backbone—see Fig. 3) (Image: 13 × 13 nm, 30 pA, 500 mV). **f** The characteristic modification of the individual molecules after annealing at higher temperature: this is reflected in a lower peak-peak corrugation in STM (see also STM profiles in Supplementary Figs. 7 & 8) and in the formation of low electron Density of States (DOS) regions at their center is consistent with dehydrogenation/cyclodehydrogenation (see main text and Fig. 3c, f) (Image: 4 × 4 nm, 30 pA, 500 mV). **g–i** N1s XP spectra of **2** on Cu(111): **g** as-deposited as in **a**, **d**; **h** after annealing at 150 °C as in **c**, **c**; **i** after annealing at 300 °C as in **c**, **f**. For the corresponding data for **1** see Supplementary Fig. 6.

the line structure, followed by cyclodehydrogenation to **2**-ox$_2$ (**1**-ox$_2$), a highly planar pyrazinacene species, which does not self-assemble, resulting in elimination of the **2**-ox (**1**-ox) line structure.

It is important to note that our data, our assignments and arguments identify that **1** and **2** behave similarly. As shown in Fig. 4 for **2**, **2**-ox and **2**-ox$_2$, (see also Supplementary Fig. 15), all these processes and structural assignments are supported by DFT simulations (Fig. 4 and Supplementary Fig. 15), on-surface profiles of the molecules (Supplementary Figs. 7, 8 and 13) and XPS data (Fig. 2g–i and Supplementary Fig. 6g–i), which not only accurately reproduce the main structural features but also correspond to the chemical changes occurring on the Cu(111) surface. That is, adsorbed **1** clearly exhibits two XPS lines assigned to unsaturated pyridine-type N atoms and amine N–H-type atoms[31,32]. This is consistent with our assignment of the

chemical process after heating to 300 °C since only pyridinic N atoms are present following this step. Cyclodehydrogenation induced by annealing at higher temperatures induces only minor changes since planarization of the compound and its modified adsorption mode do not have such a significant effect on the N shell electrons as does the change in state from amine to pyridine-type N atom. In-depth analysis of the charges on the N atoms from DFT calculations further supports this interpretation (see Supplementary Fig. 16 and Supplementary Note 2).

The DFT structures of the three closely related acene species (Fig. 4) reveal the complexity of molecular adsorption based on the oxidation state: **2** is primarily adsorbed via the terminal phenyl groups interacting in a London-type (π-metal) interaction with the metal substrate. In contrast, in **2**-ox the molecular backbone is twisted to accommodate interactions between the substrate and the pyridinic N electron pairs on the acene. Note

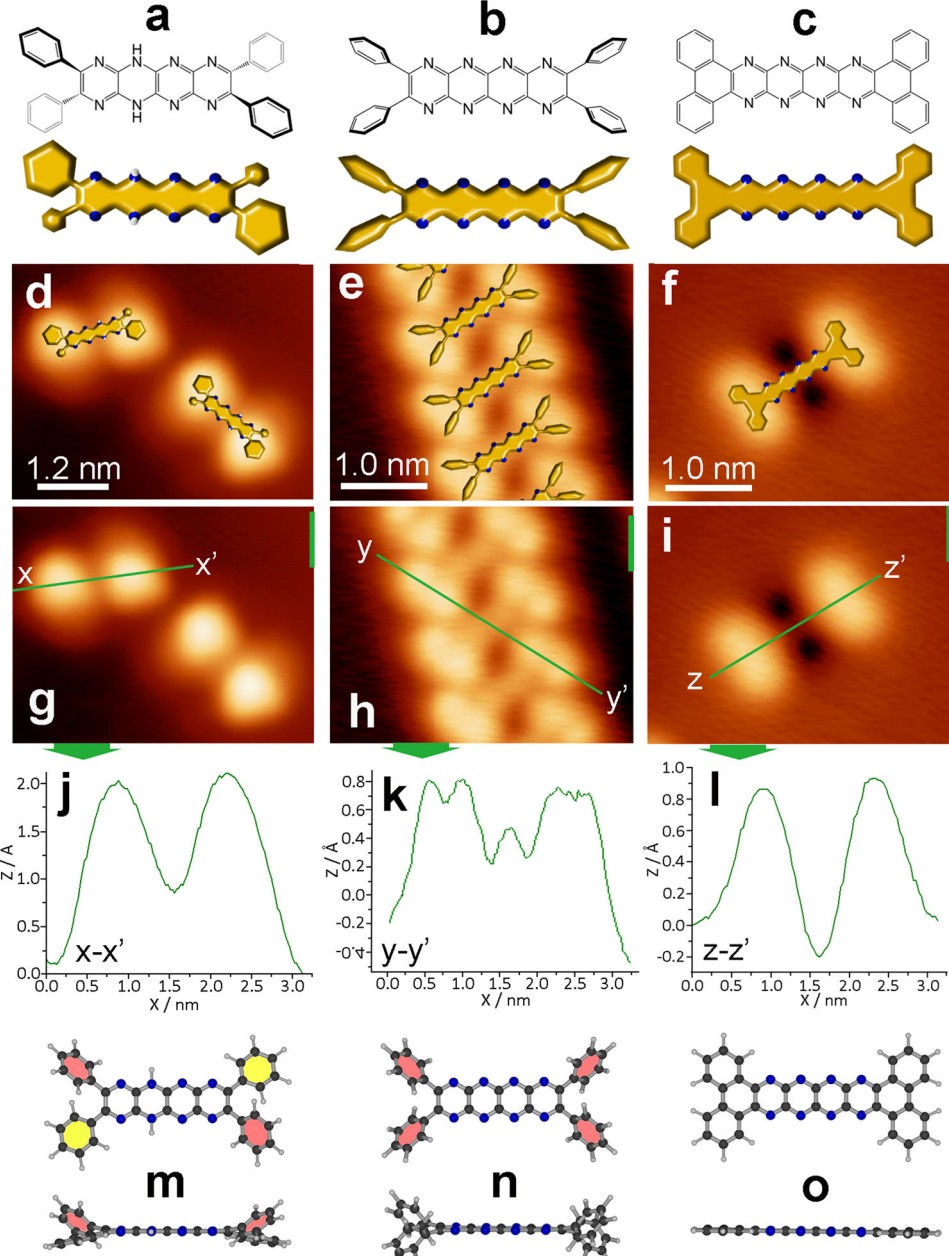

**Fig. 3 Summary of chemical species observed in the STM micrographs of 2.** Line drawing and model representation of **a 2**, **b 2**-ox, **c 2**-ox$_2$. STM images overlaid with the models of **d 2**, **e** line structure of **2**-ox, **f 2**-ox$_2$. STM images of **g 2**, **h 2**-ox, **i 2**-ox$_2$ with accompanying profile data shown respectively in **j–l**. STM profiles are consistent with chemical structures given in **m 2**, **n 2**-ox, **o 2**-ox$_2$ where different z-displacements can be associated with different dihedral angles between phenyl substituents and pyrazinacene backbone (solid pink shading—high phenyl dihedral angle ~50°; solid yellow shading—low phenyl dihedral angle ~10°). In **o**, cyclodehydrogenation at phenyl ortho positions leads to molecular planarization and an overall low STM profile. Line structures of **1**-ox have additional features which are discussed in Supplementary Figs. 13 & 14.

that acenes are known to be able to accommodate significant twisting of the backbone[34]. Thus, the modification of the interaction of the nitrogen atoms is plausibly weak and essentially undetectable by XPS in spite of a slight broadening (see also Fig. 2g–i and Supplementary Fig. 6g–i). Also note that this interaction does not arrest the lateral movement of molecules on the surface, which is required to form the self-assembled chain structures. This configuration appears to be the prerequisite for the observed cooperative conformational adaptation and aggregation into the chain form (Fig. 2). This process is favored since intermolecular C–H...N interactions within the chains allow the relaxation of twisting of the acene backbone. Following

cyclodehydrogenation, the adsorption is close to planar with the feature that the center of the backbone more closely approaches the surface in agreement with an earlier report on pentacene[35].

**Solution state properties.** The compounds **1**-ox, and **2**-ox generated in the on-surface studies are important N-heteroacene derivatives and their solution state chemical properties are also of interest especially with regard to their possible availabilities for wider study and applications. The oxidation and reduction of **1**, **2**, **1**-ox and **2**-ox has been explored by UV/Vis and fluorescence spectroscopy as shown in Fig. 5a, b together with photographic

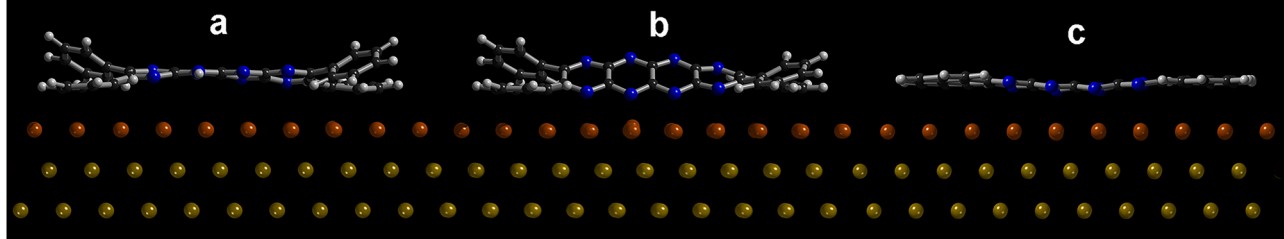

**Fig. 4 DFT calculated structures of 2, 2-ox and 2-ox₂. a** Lateral view of the native molecule **2**; diagonally-opposing phenyl rings and the main body of the molecule are oriented parallel to the surface. **b** Lateral view of the molecule which forms lines **2-ox**; pyrazinacene backbone undergoes a torsional twist enabling N atoms at one side of the acene body to interact with the Cu(111) surface. Diagonally opposed phenyl rings again interact with the surface. **c** Lateral view of the cyclodehydrogenated molecule (**2-ox₂**). The fully oxidized compound lies essentially flat with the shortest distance between molecule and surface compared to the other compounds as is also confirmed by the STM profile analysis and DFT-simulated STM images shown in Supplementary Fig. 15.

**Fig. 5 Redox state identification of pyrazinacenes 1 and 2 in the solution phase. a** UV/Vis (solid lines) and fluorescence emission spectra (dashed lines) of **1** and **2** in dry CH₂Cl₂. Insets show photographs of solutions under room light and 365 nm ultraviolet light. **b** UV/Vis (solid lines) and fluorescence emission spectra (dashed lines) of **1-ox** and **2-ox** prepared by respective treatment of **1** and **2** with PbO₂ in CH₂Cl₂. Insets show photographs of the respective solutions under room light. **c** Spectroelectrochemistry of (left) **2** and (right) **1** in benzonitrile with applied positive potential of 1.0 V. Insets of **c** show the respective differential pulsed voltammograms. **d** Thermally induced oxidation in air: variation in solid state UV/Vis spectrum of **1** prior to (orange line) and following (blue line) heating in air at 370 °C. Insets show photographs of the corresponding solid states. **e** Photoreduction of **1-ox** in CH₂Cl₂ promoted by irradiation from a Hg/Xe lamp; the UV/Vis spectrum of the reduced state of **1** is re-emerging. Blue arrows denote the decreasing intensity of the peaks corresponding to the oxidized form **1-ox**, orange arrows denote the emerging intensity of the peak associated with the native form, **1**.

images of their solutions in dichloromethane (CH$_2$Cl$_2$). **1** shows an acene-type 4-banded spectrum with an absorption maximum at 500 nm and shorter wavelength peaks diminishing in intensity. **2** has a similar several banded spectrum except that the band at 450 nm (the 2nd acene band) is the absorption maximum; a feature we attribute to tautomerization in this molecule and which has been observed for fluoflavine (dihydro-2,3,8,9-tetra-azatetracene)[36] earlier. **1**-ox and **2**-ox were obtained by treating solutions of **1** or **2** with PbO$_2$[37] both undergoing a shift in absorption maximum to longer wavelength (590 nm for **1**-ox and 540 nm for **2**-ox; see Fig. 5b). The changes in structure of the electronic absorption spectra upon PbO$_2$ oxidation of **1** and **2** are very similar to those observed for other substituted fluorubine derivatives[38]. Oxidation is also associated with substantial attenuation of fluorescence although emission can still be observed (Fig. 5b). Quantum yields of fluorescence $\Phi_F$ for the dihydro compounds **1** and **2** are, respectively, 0.88 and 0.78 in CH$_2$Cl$_2$ solution while $\Phi_F$ for both oxidized compounds is around 0.01. Interestingly, $\Phi_F$ for both **1** and **2** is also attenuated in the presence of a base (triethylamine; $\Phi_F$(**1**) = 0.64, $\Phi_F$(**2**) = 0.31) or acid (trifluoroacetic acid; $\Phi_F$(**1**) = 0.26, $\Phi_F$(**2**) = 0.15) revealing that these compounds could be sensitive reversible probes of acidity/basicity or redox conditions based on changes in their fluorescence emission.

Oxidation of **1** and **2** can also be achieved electrochemically (see Fig. 5c) with **2** being easier to oxidize than **1** by around 300 mV. This can be attributed to the greater electron deficiency of **1** over **2**. Heating **1** or **2** in air also leads to changes in UV/Vis spectra consistent with oxidation to the **1**-ox and **2**-ox states (Fig. 5d for **1**; see Supplementary Fig. 17 for **2**) with the distinct blue colour of **1**-ox emerging during heating at temperatures above 400 °C (see Supplementary Figures 18 & 19 for further details). Thermogravimetric analyses (Supplementary Figs. 18–20) reveal that the compounds strongly sequester solvents (for **1**, two molecules of water are bound while for **2**, two molecules of tetrahydrofuran are H-bonded as shown by the X-ray crystal structure). Heating either of the compounds leads to colour changes to those of their oxidized states (accompanied by UV-vis changes consistent with those). TGA of **1** indicates the excellent stability of this compound (albeit in its oxidized state) up to over 400 °C and supports the assignment of **1**-ox in the on-surface studies.

Solutions of **1**-ox and **2**-ox are sufficiently stable to allow the measurement of their UV/Vis spectra (Fig. 5b) but gradually return to their parent reduced states on standing. This process is accelerated by irradiation of solutions with UV light (Fig. 5e) with the respective characteristic spectra of the parent dihydro compounds **1** and **2** gradually re-emerging. Compound **2**-ox was found to have a low reduction potential of –0.29 mV (see Supplementary Fig. 21) also suggesting its facile reduction. Furthermore, deprotonation of **1** and **2** was found not to promote oxidation (by aerial oxidation) revealing the relative stability even of their anions in the solution state (see Supplementary Figs. 4 and 5). It should also be mentioned here that our solution-based attempts to further oxidize **1**-ox and **2**-ox by cyclodehydrogenation at the terminating diphenylpyrazine units using anhydrous FeCl$_3$ or VOF$_3$ to yield the higher acene compounds **1**-ox$_2$ and **2**-ox$_2$ have not succeeded. These investigations yielded only intractable products that could not be purified possibly due to formation of insoluble polymers. The use of appropriately substituted monomers might improve matters in this respect and we are currently investigating this matter.

## Discussion

Electronic structures of acene compounds are important for their reactivity and properties. Thus, DFT has been used to calculate their highest occupied molecular orbitals and lowest unoccupied molecular orbitals (HOMO/LUMO) (Supplementary Figs. 2 and 3 and 22 and 23). The HOMO structures of **1** and **2** are similar to those found for other pyrazinacenes[19,39] and reduced N-heteroacenes such as fluorubine[40] and dihydrodiazapentacene[41]. However, oxidation differentiates the two compounds: while **2**-ox attains an acene-like form for its molecular orbitals, **1**-ox gains a delocalized state due to the presence of nitrogen lone pairs on adjacent pyrazine units (Supplementary Fig. 23c). It appears that structures containing five or more fused pyrazine rings prefer delocalization of the pyrazine nitrogen atom lone pairs which, together with our broad experimental study, sheds new light on the DFT property predictions of Winkler and Houk[42]. The limited stability, i.e. the propensity to be reduced of **1**-ox (and to a lesser extent **2**-ox) mirrors the relative instability of pentacene. This might be related to the electron deficiency of DAP and indicates that special treatment (i.e. to exclude reducing species) will be required in investigations of the properties of these oxidized states. Photolytic redox transformations of **1** and **2** also suggest their potential as photocatalysts (see Supplementary Note 3, Supplementary Scheme 1 & Supplementary Table 1).

## Conclusions

In this unique side-by-side investigation using different complementary techniques, we report the synthesis, the physicochemical properties and in particular the oxidation-state-dependent self-assembly of pyrazinacenes. Dihydro-octaazatetracene (**2**) and dihydro-decaazapentacene (**1**), i.e. the 8- and 10-nitrogen atom-substituted analogues of tetracene and pentacene, have been selected to represent these acenes. They are shown to be oxidized. respectively. to octaazatetracene (**2**-ox) and decaazapentacene (**1**-ox) both in-solution and in situ on a solution-free surface in a vacuum in a scanning tunneling microscope. The oxidized molecules aggregate with specific supramolecular chain motifs. Onward temperature-induced cyclodehydrogenation of their phenyl substituents creates their planar analogues, **1**-ox$_2$ and **2**-ox$_2$, respectively. These adsorb as individual molecules exhibiting distinct conformations indicating that oxidation state has a critical effect on the supramolecular polymerization in this system. Our electrochemical data supports the multi-stage ionization and relatively low first oxidation potentials (and low reduction potential of the oxidized states also in-solution). In photo-stimulated redox reactions, C–C bonds are formed catalytically. For their easy dehydrogenation, in particular at interfaces, we propose that these compounds may be useful as electron/proton transfer conduits for catalysis. It is notable that **1** and **2** are both sufficiently stable to be used under reaction conditions involving irradiation where similar acene compounds may not be stable. The molecular analogues of **1**-ox and **2**-ox, pentacene and tetracene, are widely considered of most importance for their semiconductivity and their consequent use in e.g. organic electronics and optoelectronics. Decaazapentacene, on the other hand, appears to be interesting because of its photochemical behaviour as a proton-coupled chromophore that can be oxidized, deprotonated and protonated as reported here. Also, decaazapentacenes provide an example of surface-specific reactivity, i.e. of the formation of **1**-ox$_2$ and **2**-ox$_2$ exclusively on the surface, which could not be reproduced in-solution.

Based on the oxidation-state-coupled on-surface molecular morphology variations, we suggest the term 'on-surface shape-shifters' to describe these compounds. Their chemical complexity motivates further investigations comparing in-solution and interfacial reactivity, in particular towards tunable photo-redox compounds or the generation of synthetically inaccessible molecules. All these key aspects qualify the pyrazinacenes as significant members of the acene family of materials.

## Methods

**General information.** Reagents and dehydrated solvents used for synthesis and spectroscopic measurements were obtained from Aldrich Chemical Co., Tokyo Kasei Chemical Co., Wako Chemical Co. or and were used as received. Electronic absorption spectra were measured using JASCO V-570 UV/Vis/NIR spectrophotometer, Princeton Applied Research (PAR) diode array rapid scanning spectrometer or a Shimadzu UV/Visible spectrophotometer. Fluorescence spectra were measured using a JASCO FP-670 spectrofluorimeter. Fluorescence quantum yields were measured from solutions of the compounds in dichloromethane (O.D. ~0.5; c ~$10^{-5}$ M) using a Hamamatsu Photonics C9920-02G Quantum Yield Spectrophotometer. Fourier transform infrared spectra were obtained from samples deposited on a barium fluoride disc using a Thermo-Nicolet 760X FT-IR spectrophotometer. Proton and carbon-13 nuclear magnetic resonance ($^1$H-NMR and $^{13}$C NMR) spectra were obtained using a JEOL AL300BX spectrometer ($^1$H: 300 MHz; $^{13}$C: 75 MHz) or a Bruker Avance III 600 MHz spectrometer ($^1$H: 600 MHz; $^{13}$C: 150 MHz) with tetramethylsilane (TMS) as an internal standard. MALDI-TOF mass spectra were obtained using a Shimadzu Axima AFR+ mass spectrometer using dithranol as the matrix. High-resolution (HR) MALDI-TOF mass spectra were measured using an LTQ Orbitrap XL system using 2,5-dihydroxybenzoic acid as matrix. Computational geometry optimizations were performed using NWChem[43] at the B3LYP/6-311 + G(d,p) level. GaussView (in GAUSSIAN[44]) was used to generate images of frontier HOMO and LUMO orbitals. Cyclic voltammograms were recorded on an EG&G Model 263 A potentiostat operating a three-electrode system. The working electrode was a platinum button with a platinum wire serving as the counter electrode. An Ag/AgCl electrode was used as the reference. Tetra-n-butylammonium perchlorate (0.2 M) was used as the electrolyte and voltammograms were recorded at a scan rate of 100 mV s$^{-1}$. Ferrocene/ferrocenium redox couple was used as internal standard and all solutions were purged using argon gas prior to electrochemical/spectral measurements. Spectroelectrochemical study was performed by using a cell assembly (SEC-C) supplied by ALS Co., Ltd. (Tokyo, Japan). This assembly consists of a Pt counter electrode, a 6 mm Pt gauze working electrode, and an Ag/AgCl reference electrode in a 1.0 mm path length quartz cell. Optical transmission was limited to 6 mm covering the Pt gauze working electrode. 5,6-Diphenyl-2,3-diaminopyrazine (**6**) was prepared according to a reported procedure[45]. Syntheses of 2,3-dichloro-6,7-diphenyl-1,4,5,8-tetraazanaphthalene, 2,3-diamino-6,7-diphenyl-1,4,5,8-tetra-azanaphthalene, 2,3-dichloro-6,7-di(4-t-butylphenyl)-1,4,5,8-tetraazanaphthalene and 2,3-diamino-6,7-di(4-tert-butylphenyl)-1,4,5,8-tetraazanaphthalene are given in Supplementary Methods in the Supplementary Information.

## Synthesis

*5,12-Dihydro-2,3,8,9-tetraphenyl-1,4,5,6,7,10,11,12-octaazatetracene, 2.* A mixture of 2,3-dichloro-6,7-diphenyl-1,4,5,8-tetraazanaphthalene (106 mg, 0.3 mmol), 2,3-diamino-5,6-diphenylpyrazine (95 mg, 0.36 mmol) and K$_2$CO$_3$ (138 mg, 1 mmol) in DMSO (5 mL) was heated for 3 h at 110 °C. The reaction mixture was allowed to cool to room temperature then poured into saturated aqueous ammonium chloride solution (30 mL) and extracted with chloroform (3 × 50 mL). The combined extracts were washed with brine (100 mL), dried over anhydrous Na$_2$SO$_4$ then solvents evaporated under reduced pressure. The product was isolated using column chromatography (SiO$_2$; CHCl$_3$, 8% THF) to give the product as an orange crystalline solid. Yield: 70 mg (43 %). $^1$H-NMR (300 MHz, THF-d8, 25 °C) $\delta$ = 7.14–7.22 (m, 12H, ArH), 7.33–7.36 (m, 8H, ArH), 10.94 (br. s, 2H, NH) ppm. $^{13}$C NMR (75 MHz, CF$_3$COOD, 25 °C) $\delta$ = 131.3, 131.5, 133.2, 134.1, 142.3, 149.3, 152.0 ppm. FT-IR (BaF$_2$) $\nu$ = 3639, 2956, 2924, 2854, 1522, 1445, 1429, 1406, 1368, 1185, 1088, 1026, 984, 762, 695 cm$^{-1}$. HR-MS (MALDI, DHB): m/z calc. for C$_{34}$H$_{22}$N$_8$ [M + 2H]$^+$: 544.21184; found 544.21385. UV-Vis (CH$_2$Cl$_2$): $\lambda_{max}$ ($\varepsilon$) = 249 (77000), 441 (39500), 461 (67000), 489 (46000) nm. Elemental analyses: calc'd for C$_{34}$H$_{24}$N$_8$.2(C$_4$H$_8$O).1/2(C$_6$H$_{14}$): %C 74.05, %H 6.21, %N 15.35; found %C 74.51, %H 5.68, %N 15.55.

*6,13-Dihydro-2,3,9,10-tetraphenyl-1,4,5,6,7,8,11,12,13,14-decaazapentacene, 1.* A mixture of 2,3-dichloro-6,7-diphenyl-1,4,5,8-tetraazanaphthalene (106 mg, 0.3 mmol), 2,3-diamino-6,7-diphenyl-1,4,5,8-tetraazanaphthalene (104 mg, 0.33 mmol) and K$_2$CO$_3$ (138 mg, 1 mmol) in DMSO (5 mL) was heated for 4 h at 120 °C. The reaction mixture was allowed to cool to room temperature then poured into aqueous saturated ammonium chloride solution (30 mL) and filtered. The filtrate was dissolved in THF, the resulting solution passed through a short plug of silica then the solvents were evaporated under reduced pressure. Ethyl acetate (20 mL) was added to the resulting solid followed by ultrasonication of the suspension for 10 min. The suspension was filtered, rinsed with water (50 mL), hot ethyl acetate (10 mL), dichloromethane (10 mL) and dried under reduced pressure to give the product as a red crystalline solid. Yield: 63 mg (35 %). UV-Vis (CH$_2$Cl$_2$): $\lambda_{max}$ ($\varepsilon$) = 252 (75,100), 470 (73,500), 502 (108,000) nm. $^1$H-NMR (300 MHz, CF$_3$COOD, 25 °C) $\delta$ = 7.05–7.08 (m, 8H, ArH), 7.15–7.17 (m, 8H, ArH). $^{13}$C NMR (75 MHz, CF$_3$COOD, 25 °C) $\delta$ = 131.6, 132.2, 133.5, 135.4, 143.0, 147.9, 155.6 ppm. FT-IR (BaF$_2$): $\nu$ = 3631, 3384, 3085, 2948, 1580, 1507, 1445, 1192, 1089, 1026, 696 cm$^{-1}$. HR-MS (MALDI, DHB): m/z calc. for C$_{36}$H$_{22}$N$_{10}$ [M–H]$^-$: 593.19561; found 593.19331. Elemental analyses: calc'd for C$_{34}$H$_{23}$N$_{10}$.2.5H$_2$O: %C 67.60, %H 4.25, %N 21.90; found %C 67.82, %H 4.06, %N 21.63.

*6,13-Dihydro-2,3,9,10-tetrakis(4-t-butylphenyl)-1,4,5,6,7,8,11,12,13,14-decaazapentacene, 1-tBu.* A mixture of 2,3-dichloro-6,7-bis(4-tert-butylphenyl)-1,4,5,8-tetra-azanaphthalene (140 mg, 0.3 mmol), 2,3-diamino-6,7-bis(4-tert-butylphenyl)-1,4,5,8-tetraazanaphthalene (141 mg, 0.33 mmol) and K$_2$CO$_3$ (138 mg, 1 mmol) in DMSO (5 mL) was heated for 4 h at 120 °C. The reaction mixture was allowed to cool to room temperature, poured into aqueous saturated ammonium chloride solution (30 mL) then the mixture extracted with CHCl$_3$ (3 × 50 mL). The combined extracts were washed with brine (50 mL), dried over anhydrous Na$_2$SO$_4$ then solvents were evaporated under reduced pressure. The solid residue was purified using column chromatography (SiO$_2$; CHCl$_3$, 8% THF) to give the product as a dark purple crystalline solid. Yield: 110 mg (45 %). UV-Vis: $\lambda_{max}$ ($\varepsilon$) = 231 (31,400), 268 (22,800), 571 (51,700), 606 (60,500) nm. $^1$H-NMR (300 MHz, THF-d$_8$, 25 °C): $\delta$ = 7.25–7.30 (m, 8H, ArH), 7.40–7.45 (m, 8H, ArH), 11.61 (br. s, 2H, NH) ppm. $^{13}$C NMR (75 MHz, CF$_3$COOD, 25 °C) $\delta$ = 31.7, 36.9, 128.5, 130.2, 131.8, 142.37, 147.3, 155.5, 160.8 ppm. FT-IR (BaF$_2$) $\nu$ = 3213, 3129, 3051, 2960, 2924, 2855, 1607, 1542, 1442, 1286, 1190, 1116, 1077, 1016 cm$^{-1}$. HR-MS (MALDI, DHB): m/z calc. for C$_{52}$H$_{54}$N$_{10}$ [M – H]$^-$: 817.44601; found 817.44292.

*General procedure for oxidation.* Dihydroazaacene derivative (0.001 mmol) was dissolved in dry dichloromethane (3 mL) and lead (IV) oxide (50 wt%) was added. The reaction mixture was stirred for 10 min. then filtered through Celite and solvents evaporated under reduced pressure to give the products as solids in quantitative yields.

*2,3,8,9-Tetraphenyl-1,4,5,6,7,10,11,12-octaazatetracene, 2-ox.* Dark purple solid. $^1$H-NMR (300 MHz, CD$_2$Cl$_2$, 25 °C): $\delta$ = 7.45–7.50 (m, 8H, ArH), 7.56–7.61 (m, 8H, ArH), 7.82–7.86 (m, 8H, ArH) ppm. FT-IR (BaF$_2$) $\nu$ = 3051, 2916, 2848, 1593, 1540, 1427, 1366, 1130, 768, 698, 693 cm$^{-1}$. HR-MALDI-MS (DHB): m/z [M + H]$^+$ calcd for C$_{34}$H$_{21}$N$_8$: 541.1889; found 541.1888.

*2,3,9,10-Tetraphenyl-1,4,5,6,7,8,11,12,13,14-decaazapentacene, 1-ox.* Dark red solid. FT-IR (BaF$_2$) $\nu$ = 2947, 2865, 1580, 1539, 1445, 1192, 1089, 1070, 697 cm$^{-1}$. HR-MALDI-MS (DHB): m/z [M + H]$^-$ calcd for C$_{36}$H$_{21}$N$_{10}$: 593.1951; found 593.2013. This compound is too insoluble to perform solution state NMR analyses.

*2,3,9,10-Tetrakis(4-t-butylphenyl)-1,4,5,6,7,8,11,12,13,14-decaazapentacene, 1-tBu-ox.* Dark purple solid. $^1$H-NMR (300 MHz, CD$_2$Cl$_2$, 25 °C): $\delta$ = 1.40 (s, 36H, CH$_3$), 7.52 (d, $^3J$ = 9 Hz, 8H, ArH), 7.88 (d, $^3J$ = 9 Hz, 8H, ArH) ppm. $^{13}$C NMR (75 MHz, CD$_2$Cl$_2$, 25 °C) $\delta$ = 30.9, 35.2, 125.8, 130.9, 134.6, 148.6, 148.7, 156.8, 163.8 ppm. FT-IR (BaF$_2$) $\nu$ = 3067, 2962, 2904, 2868, 1603, 1505, 1407, 1340, 1304, 1196, 1097, 1062, 831, 704 cm$^{-1}$. HR-MALDI-MS (DHB): m/z [M + H]$^-$ calcd for C$_{52}$H$_{53}$N$_{10}$: 817.4460; found 817.4432.

**X-ray crystallography.** Crystals of **2** were grown by diffusion of hexane into a solution of **2** in tetrahydrofuran. Data collections were performed using MoK$_\alpha$ radiation ($\lambda$ = 0.71073 Å) on a RIGAKU VariMax Saturn diffractometer equipped with a CCD detector. Prior to the diffraction experiment the crystals were flash-cooled to 213 K in a stream of cold N$_2$ gas. Cell refinements and data reductions were carried out by using the d*trek program package in the CrystalClear software suite[46]. The structures were solved using a dual-space algorithm method (SHELXT)[47] and refined by full-matrix least squares on F2 using SHELXL-2014[48] in the WinGX program package[49]. Non-hydrogen atoms were anisotropically refined and hydrogen atoms were placed on calculated positions with temperature factors fixed at 1.2 times U$_{eq}$ of the parent atoms and 1.5 times U$_{eq}$ for methyl groups. Crystal data for **2**: orange bar, C$_{42}$H$_{34}$N$_8$O$_2$, M$_r$ = 682.77, monoclinic P21/n, a = 15.6972(4) Å, b = 5.71170(10) Å, c = 19.7826(6) Å, $\beta$ = 106.4810(10)°, V = 1700.79(7) Å$^3$, T = 213 K, Z = 2, R$_{int}$ = 0.0428, GoF = 1.063, R$_1$ = 0.0793, wR(all data) = 0.2500.

**Scanning tunneling microscopy.** All experiments were performed with an Omicron low temperature STM operating at 4.8 K in ultrahigh vacuum (UHV). A clean Cu(111) surface was prepared in situ by repeated cycles of sputtering and annealing. Mechanically sharpened and sputter-etch cleaned PtIr tips were used. Molecules were deposited from a Knudsen cell by resistively heating at about 350 °C for both **1** and **2**. All sample preparations were performed under ultrahigh vacuum (UHV) conditions with a base pressure of ~$10^{-10}$ mbar. The crystals were cleaned using cycles of sputtering with Ar+ ions with subsequent annealing at 450 °C. Thermal evaporation at 300 °C was used to deposit the molecules on the substrate using a commercial evaporator (Kentax GmbH, Germany). The sublimed amounts of the compounds were controlled by using a quartz crystal microbalance. STM imaging was performed in constant current mode (typical tunneling current 10–50 pA) with the selected sample bias in the range of 3 mV to 1 V at 5 K. Pt-Ir wires (90% Pt, 10% Ir) were used to make scanning probe tips and these were cleaned by sputtering with Ar$^+$ ions prior to use.

**In situ X-ray photoelectron spectroscopy.** XPS was used at room temperature to track the N1s chemical environment changes of the compounds before and after annealing steps. The spectra were recorded in normal emission with an

instrumental setup that gives a full width at half maximum (FWHM) of 1.0 eV using a monochromatized Al Kα X-ray source.

**DFT simulation of on-surface structures**. Simulations were performed using the Siesta code (See the Siesta distribution site, https://launchpad.net/siesta and refs. [50,51]). This uses norm-conserving pseudopotentials[52] and expands the wave functions of valence electrons in LCAO form. As exchange-correlation functional, we have used the vdW-DF-cx functional of Berland and Hyldgaard (BH)[53]. The geometric models were constructed based on the experimental bulk parameter for copper, 3.62 Å since the BH is known to reproduce well bulk properties of coinage metals. The systems were confined to unit cells that allow the study of periodically repeating Cu(111) surface, and included three atomic layers. The super-cell sizes vary from 7 × 10 Cu atoms for monomers to 12 × 12 Cu atoms for dimers. The length of the cells along the OZ axis was 30 Å for all systems, thus allowing a vacuum level of ~20 Å, which is large enough to avoid the artificial influence of the electric charge from one cell to another. The Monkhorst-Pack grid for the integrals in the Brillouin zone was 2 × 2 × 1. We used double-zeta polarized (DZP) basis sets with an energy shift of 100 meV for all atoms. The relaxation procedure involves the Conjugated Gradient method. The systems were relaxed by keeping the deeper layer of the copper substrate pinned at their bulk positions, only the upper layer and the molecule being permitted to relax until the maximum gradient in the structure was below 0.05 eV/A. Simulations of the STM images were performed by using the Tersoff-Hamann approximation (i.e. STM image is determined by the local density of states (LDOS)[54]. We computed LDOS in the energy window spanning from the Fermi level of the system up to 1 eV below the Fermi level, corresponding to an external bias of 1 V. We scanned the surface by searching for a given constant value of the LDOS (i.e. we simulate a "constant current" STM experiment).

**Data analysis**. Experimental and simulated STM data were processed using the WSXM software[55].

## Data availability

[1]H & [13]C NMR spectra and high-resolution mass spectra of the compounds are shown in Supplementary Figs. 24–52. Crystallographic data (excluding structure factors) have been deposited with the Cambridge Crystallographic Data Centre with CCDC reference number 1576267 (**2**). Copies of the data can be obtained, free of charge, on application to CCDC, 12 Union Road, Cambridge CB2 1EZ, UK http://www.ccdc.cam.ac.uk/perl/catreq/catreq.cgi, e-mail: data_request@ ccdc.cam.ac.uk, or fax: +44 1223 336033. CIF file for compound **2** is supplied as Supplementary Data 1. Atomic coordinates of optimized structures are supplied as Supplementary Data 2–11.

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

## Acknowledgements

This work was partly supported by World Premier International Research Center Initiative (WPI Initiative), MEXT, Japan. Prof. Masakazu Aono, MANA directorate, and MANA visiting scientist program are acknowledged for their decisive role in triggering this collaboration. R. Schelldorfer and M. Martina are acknowledged for their technical support; T. Nijs for helpful discussions. This study was supported by JSPS KAKENHI Grant Numbers JP15K13684 (Linear acene proton conductors for molecular electronics). We are also grateful to the Paul Scherrer Institute, the Physics Department of the University of Basel and the Swiss Nanoscience Institute, the Swiss National Science Foundation (Grant # 200020_162512, 206021_144991, 206021_121461), the Swiss Commission for Technology and Innovation (CTI, 16465.1 PFNM-NM) and the Swiss Government Excellence Scholarship Program for Foreign Scholars. XPS measurements were performed at the Laboratory for Micro- and Nanotechnology at the Paul Scherrer Institute (PSI). L.B.-I. and C.M. acknowledge financial support from MCI Romania, CORE program, project PN19 35 02 01. We are grateful to the elemental analysis service at the University of Tsukuba for combustion analyses.

## Author contributions

D.M. synthesized and characterized the molecules with assistance from P.S., G.J.R. and J.P.H. for precursor synthesis. S.F.M., A.A. and T.A.J. performed scanning tunneling microscopy and X-ray photoelectron spectroscopy. Z.B. and F.B. measured photo-catalytic properties of the compounds. A.M. and F.D. performed and interpreted electrochemical measurements. Y.M. collected and refined single crystal X-ray data. J.L. collected and analyzed NMR and electronic spectrophotometric data. L.B.-I., P.A.K. and C.M. performed and interpreted DFT calculations on the molecules and surface-adsorbed structures. T.M. performed preliminary catalysis investigations of the compounds. Y.W. and T.A.J. performed preliminary STM measurements. J.P.H., T.A.J. and K.A. designed the experiments and supervised the work. All authors contributed to the final data analysis and preparation, and the writing of the manuscript.

## Competing interests

The authors declare no competing interests.
