## [Peer Review File · Communications Chemistry]

Reviewers' comments:

Reviewer #1 (Remarks to the Author):

This is an excellent manuscript that describes frontier work at organic synthesis and materials science. The authors describe the synthesis of pyrazine-based acenes and disclose a number of properties including catalysis. The manuscript is not just specialized and deserves credit for pushing the envelope in the the respective areas. I have no doubt that the work will garner attention. It is unclear whether the molecules will gain widespread adoption; however, the idea is set and extremely encouraging.

Reviewer #2 (Remarks to the Author):

In this research, the authors reported the synthesis, the physicochemical properties and in particular the oxidation-state dependent self-assembly of pyrazinacenes through the side-by-side investigation with different complementary techniques. The authors found that Dihydrooctaazatetracene (2) and dihydrodecaazapentacene (1) could be oxidized respectively to 348 octaazatetracene (2-ox) and decaazapentacene (1-ox) both in solution and in situ on a solution free surface in a vacuum in a scanning tunneling microscope. The electrochemical data suggested the multistage ionization and relatively low first oxidation potentials (and low reduction potential of the oxidized states also in solution). Specially, in photo-stimulated redox reactions, C-C bonds are formed catalytically. In addition, the authors also demonstrated that decaazapentacenes could provide an example of surface-specific reactivity. The research is meaningful and I would like to recommend its publication after minor revisions.

(1) There might be some mistakes in Graphical Abstract. In STM images, there are only eight Nitrogen atoms in four linearly-fused aromatic rings, however, in the given molecular structures, the authors provided the frameworks with ten nitrogen atoms in five linearly-fused rings. they did not match each other.

(2) What are the fluorescent quantum yield of 1, 2, 1-OX and 2-OX?

(3) Scanning speed and electrolyte should be provided in the caption of Figure S21

(4) Since the authors conducted on-surface chemistry at different temperature, TGA analysis of 1, 2, 1-OX and 2-OX should be performed and provided in the revised SI.

(5) Some related references should be included in the revised manuscript:

10.1039/C7CC03898D; 10.1039/C9QM00656G; 10.1002/TCR.201600015; 10.1039/C6TB02052F; 10.1002/ASIA.201300208.

Reviewer #3 (Remarks to the Author):

The manuscript entitled "Pyrazinacenes: On-surface Shapeshifters and Configurable Photoredox Catalysts" presents an interesting contribution about the control of the oxidation states of on-surface synthesized planar benzo-substituted.

The results are interesting and the experimental data and theoretical calculations strong enough to assure publication. However, some parts of the article, mainly the discussion and focus, have to be addressed before it can be further considered for publication.

In general, the presentation is a bit confusing, as the chemical part is very disconnected for the on-surface part. In fact, albeit the efforts of the authors, the manuscript involve two independent contributions (solution chemistry and on-surface chemistry). As it is, the direct comparison of both routes is not providing information to the main body of the paper but confusion.

Authors shall consider the limitations of on-surface chemistry routes to grow thick films of the different species generated on surface, that preclude any photoredox reaction. Moreover, the synthetic pathways of both methodologies are very different, and therefore is difficult to "export"

conclusions. Said that, I still have the opinion that the work merits publication but authors shall focus better the problem, for instance to stress the fact that on-surface route is able to produce species (1,2-ox2) that not reachable by standard synthesis.

Taking this into account, I suggest to the authors focusing the manuscript in the on-surface aspect of the work removing some discussion from the main text to the supplementary (or to another future work focused on induced chemical photoredox reactions).

Some specific points:

1./ title: following the previous discussion, the title does not reflect the main message. It shall be focused on the sequential on-surface synthesis of ...

2./ XPS: The XPS spectra is one important issue, as it confirms the oxidation process within the atomistic model derived from DFT. Although it is true that spectra are not of good quality, both components can be clearly seen. The spectra of fig S6 shall be promoted to the main text (fig. 2).

2.1/The XPS figure has to be improved, as the binding energy scale is not readable. Moreover, energy references shall be given in the text to prove the peak assignation.

2.2/It is surprising that the binding energy of the main peak does not change from structure 2-ox and 2-ox2, however, it does for 2. This shall be related to charge transfer process. Is that correct at the light of DFT calculations?

3./ most of the discussion is related to the relationship between oxidation state and structural conformation. In this sense, it would be more interesting to move Fig. 4 to the supplementary (does not provide any information and STM-DFT calculations are always tricky) and replace it by fig. S17. When the structure of the different molecules can be seen.

3.1/ with respect to this figure S17, I understand that calculations were performed using three substrate layers. Is this thickness enough to ensure a good structure determination?

3.2/ It is surprising that 2-ox is a bit tilted, whereas 2 is not. Is any chemical bond between the N atoms and the surface (which are the involved distances from N to the surface) ? Shall this interaction be therefore reflected in the XPS spectrum?

3.2/ An important point related with my query 2.2, is about the charge transfer from the surface, if any. Is there any specific and different charge transfer associated to each oxidation state? In literature, some molecules have been found to curve when accepting charge on the surface. Is this the case?

4./ I cannot see the relation between the photo redox studies and the synthesis on the surface. It is a completely different characterization. The important point is that these wet routes does not produce 1-ox2 and 2-ox2 species. In this sense, the photoredox catalysis paragraph can be removed from the main text.

5./ Transition from isolated molecules on the surface to ordered aligned structures is very interesting. It is not a polymerization process (line 353), and I wonder which is the role of diffusion or surface-molecule interaction for this structural transformation. Why 2 and 2-ox2 does form individual structures whereas 2-ox1 self-assemble in aligned structures, is it related with the molecular tilt? There is some pi-pi interaction between adjacent molecules?

Responses to Reviewers' comments:

Reviewer #1 (Remarks to the Author):

This is an excellent manuscript that describes frontier work at organic synthesis and materials science. The authors describe the synthesis of pyrazine-based acenes and disclose a number of properties including catalysis. The manuscript is not just specialized and deserves credit for pushing the envelope in the the respective areas. I have no doubt that the work will garner attention. It is unclear whether the molecules will gain widespread adoption; however, the idea is set and extremely encouraging.

RESPONSE: We thank the reviewer for these positive and encouraging remarks. We intend to develop the molecules and materials presented in this work as far as is possible.

Reviewer #2 (Remarks to the Author):

In this research, the authors reported the synthesis, the physicochemical properties and in particular the oxidation-state dependent self-assembly of pyrazinacenes through the side-by-side investigation with different complementary techniques. The authors found that Dihydrooctaazatetracene (2) and dihydrodecaazapentacene (1) could be oxidized respectively to 348 octaazatetracene (2-ox) and decaazapentacene (1-ox) both in solution and in situ on a solution free surface in a vacuum in a scanning tunneling microscope. The electrochemical data suggested the multistage ionization and relatively low first oxidation potentials (and low reduction potential of the oxidized states also in solution). Specially, in photo-stimulated redox reactions, C-C bonds are formed catalytically. In addition, the authors also demonstrated that decaazapentacenes could provide an example of surface-specific reactivity. The research is meaningful and I would like to recommend its publication after minor revisions.

RESPONSE: We thank the reviewer for these positive remarks.

1) There might be some mistakes in Graphical Abstract. In STM images, there are only eight Nitrogen atoms in four linearly-fused aromatic rings, however, in the given molecular structures, the authors provided the frameworks with ten nitrogen atoms in five linearly-fused rings. they did not match each other.

RESPONSE: The graphical abstract has been revised. In the original version, we had hoped to convey the use of N8 and N10 compounds. However, the reviewer is correct and we have replaced the lower part of the image with STM and model images of N10 for consistency.

Action taken: The TOC has been revised (Pg. 4).

2) What are the fluorescent quantum yield of 1, 2, 1-OX and 2-OX?

RESPONSE: Reduced compounds have rather high quantum yields and this can be varied according to the state of protonation. While the oxidized compounds do exhibit fluorescence, it is very weak compared to C-H only acenes. The reasons for this are not fully understood although we note the unusual narrow

emission profile of 1-ox. These issues are critical for the optical and electronic properties of the compounds and their applications as they are further investigated at present.

Action taken: Fluorescence quantum yields for 1, 2, 1-ox and 2-ox have been added to the manuscript and a brief discussion of the data included. Data related to the compounds in acidic and basic solution has also been added. Pg 17, Ln 15. Details about the instrumentation was used for these measurements has been included in the Methods 'General information' section Pg22 Ln 16.

3) *Scanning speed and electrolyte should be provided in the caption of Figure S21.*

RESPONSE: The figure caption has been updated. It was necessary to use distilled chloroform to collect this data due to the lower stability of these compounds against reduction. 0.2 M tetrabutylammonium perchlorate was used as the electrolyte in cyclic voltammetry at a scan rate of 100 mV s⁻¹.

Action taken: The information has been placed in the relevant Figure caption and the experimental description has been improved (Supplementary Fig. S21).

4) *Since the authors conducted on-surface chemistry at different temperature, TGA analysis of 1, 2, 1-OX and 2-OX should be performed and provided in the revised SI.*

RESPONSE: The TGA data for 1 and 2 had been contained in Supplementary Figs. S19, S20 of the original submission (now Supplementary Figs. S18, S19). Our investigations of the thermal properties of 1 and 2 reveal that they are both oxidized at higher temperatures. The oxidized compounds, 1-ox and 2-ox, respectively are formed but due to the small change in mass are not so apparent in our TGA data. Thus, the TGA measurements performed on 1 and 2 initially reflect the properties of the reduced compounds (1 and 2) and at higher temperature reflect the oxidized states (1-ox and 2-ox). This is particularly apparent for 1/1-ox where the intense blue colour (characteristic of 1-ox) can clearly be observed after heating the sample above 400 C (Fig. S19). We have collected TGA data again for temperatures above 600 C using Pt sample holder up to 850 C (originally only Al pans were available for this). This new data is shown in Supplementary Fig. S20.

Action taken: New TGA data is shown in Supplementary Fig. S20.

5) *Some related references should be included in the revised manuscript.*

RESPONSE and action taken: We have added these citations at the relevant points in the text of the main manuscript: references 17, 18, 23, 24 & 29 (not in the same order as mentioned in the review comment) and the other references renumbered accordingly.

Reviewer #3 (Remarks to the Author):

The manuscript entitled 'Pyrazinacenes: On-surface Shapeshifters and Configurable Photoredox Catalysts' presents an interesting contribution about the control of the oxidation states of on-surface synthesized planar benzo-substituted.

The results are interesting and the experimental data and theoretical calculations strong enough to assure publication. However, some parts of the article, mainly the discussion and focus, have to be addressed before it can be further considered for publication.

In general, the presentation is a bit confusing, as the chemical part is very disconnected for the on-surface part. In fact, albeit the efforts of the authors, the manuscript involve two independent contributions (solution chemistry and on-surface chemistry). As it is, the direct comparison of both routes is not providing information to the main body of the paper but confusion.

RESPONSE: It is true that the manuscript shows complementary data about the compounds. We feel strongly that the novelty of 1 and 2 and their oxidized states justifies that we also present some essential bulk/solution properties together with their interesting on-surface properties. In particular, we investigated the accessibility of the compounds in solution state after having observed the on-surface transformations. Reviewers 1 and 2 support this approach and reviewer 3 obviously also appreciates the importance of on-surface synthesis to access, for instance, the fused compounds 1-ox2 and 2-ox2 which were not accessible in the solution state (see the next review comment and response).

Action taken: To remove the less relevant text in accordance with the reviewer's suggestion, the details of the solution state photoredox catalysis have been moved to the Supplementary Information. A short statement and reference to SI has been added to the main manuscript (Pg. 20, last sentence).

Authors shall consider the limitations of on-surface chemistry routes to grow thick films of the different species generated on surface, that preclude any photoredox reaction. Moreover, the synthetic pathways of both methodologies are very different, and therefore is difficult to export conclusions. Said that, I still have the opinion that the work merits publication but authors shall focus better the problem, for instance to stress the fact that on-surface route is able to produce species (1,2-ox2) that not reachable by standard synthesis.

Taking this into account, I suggest to the authors focusing the manuscript in the on-surface aspect of the work removing some discussion from the main text to the supplementary (or to another future work focused on induced chemical photoredox reactions).

RESPONSE: We agree: Compounds 1 and 2 and the products of oxidation at dihydropyrazine units are accessible by solution methods; the other oxidation states 1-ox2 and 2-ox2 are so far only available by on-surface synthesis. In absence of an effective synthesis route of 1-ox2 and 2-ox2, the sole availability of on-surface synthesis might preclude uses of the ultimate oxidized compounds 1-ox2 and 2-ox2 for photoredox processes, as the reviewer has stated. In accordance with the reviewer's suggestion we have moved the discussion on photoredox catalysis to the supporting information, adjusted the text to reflect this revision and removed some text regarding in situ preparation of organic semiconductors.

Action taken: The passage on photoredox work and the citations therein has been shifted to ESI (as a documentation of the present state of knowledge to be expanded on later). See also response to point 4. below. Also, we have removed the statement (Conclusion, paragraph 2) about the possibility of preparing new organic semiconductors since that would require thick films of any *in situ* prepared molecules (Pg 20 & Conclusion paragraph 2).

Some specific points:

1./ title: following the previous discussion, the title does not reflect the main message. It shall be focused on the sequential on-surface synthesis of the compounds;

RESPONSE: Based on the other comments and revisions, we agree.

Action taken: title revised to better reflect the manuscript contents: "Pyrazinacenes: On-surface Redox-coupled Shapeshifters" Pg 1 Ln 1

2./ XPS: The XPS spectra is one important issue, as it confirms the oxidation process within the atomistic model derived from DFT. Although it is true that spectra are not of good quality, both components can be clearly seen. The spectra of fig S6 shall be promoted to the main text (fig. 2).

2.1/The XPS figure has to be improved, as the binding energy scale is not readable. Moreover, energy references shall be given in the text to prove the peak assignment.

Action taken: We have improved the readability of the axis labels in Figure 2. We have also added details to the text of the assignments with literature citations. (Text and citations: Pg 12, Ln 7-8; Citations: Ln 18-19 and Pg 13, Ln 21-22)

2.2/It is surprising that the binding energy of the main peak does not change from structure 2-ox and 2-ox2, however, it does for 2. This shall be related to charge transfer process. Is that correct at the light of DFT calculations?

RESPONSE (Comment 2 all parts): We have updated the XPS figure and added discussion text concerning the surface chemical analysis to the main manuscript. Also, as the reviewer suggests, a discussion of the XPS shifts between 2-ox and 2-ox2 has been added on the basis of XPS and DFT. A further in depth DFT analysis (Voronoi Charges) makes it plausible that no significant XPS shift is observed between 2-ox and 2-ox-2. A short discussion has been added to the main text (Pg 15, Ln 7-14) and a longer part to Supplementary Information together with Supplementary Fig. S16.

Action taken: Figure S10 (XPS spectra corresponding to the compounds 2, 2-ox and 2-ox2 has been moved to the main manuscript and incorporated into Fig.2.

Text added to main manuscript to accompany the XPS Data

That is, adsorbed **1** clearly exhibits two XPS lines assigned to unsaturated pyridine-type N atoms and amine N-H type atoms^{31,32}. This is consistent with our assignment of the chemical process after heating to 300 C since only pyridinic N atoms are present following this step. Cyclodehydrogenation induced by annealing at higher temperatures induces only minor changes since planarization of the compound and its modified adsorption mode do not exert a comparable effect on the N shell electrons as does the change in state from amine to pyridine-type N atom. The absence of an observable chemical shift also evidences that neither the original compound **1** and **2**, nor their oxidized forms are observed in a chemisorbed state. In-depth analysis of the charges on the N atoms from DFT calculations further supports this interpretation (Supplementary Fig. S17 and accompanying explanation). (Pg 15, Ln 7-14)

Text accompanying Supplementary Fig S16: To discuss the energies of the lines in the XPS N1s spectra we have analyzed the electronic populations of the adsorbed molecules. The Voronoi charges^{S1,S2} gained/lost for each atom are given in Supplementary Fig S17 above. In particular, it can be seen that for **2-ox** and **2-ox₂** the Voronoi populations are not related to the average distance between the nitrogens and the surface. Indeed, the geometry of the **2-ox** system indicates that the nitrogens on one side of the central part of the molecule are closer to the surface than those on the other side. The average value for the Voronoi population, however, is -0.16 in both cases (i.e. left and right side of the central pi system). For the **2ox-2** system, the average value is very close to that obtained in the case of **2-ox**, i.e. -0.17 e. While the average values are the same in both cases, differences up to 0.04 e may occur between specific atoms, most probably caused by the local geometry: the distance between a given nitrogen and the closest Cu atom will specifically influence its electronic population. Indeed, in vacuum these differences amount to less than 0.01 e in most cases. For **2**, we see that N bonded to H atoms exhibits population around -0.04 e, while for the other nitrogens exhibiting a free electron pair, the average population is -0.16 e. This agrees favorably with the observation that only the native molecules **1** and **2** exhibit two peaks. The first lower intensity signal is due to nitrogens bonded to N-H (corresponding to a population of -0.04 e) with a second one specific to other pyridine-type N. This peak is rather similar in the Voronoi charge argument regardless of the position of the N atom within the molecule and whether it is dehydrogenated or cyclodehydrogenated. This peak corresponds to an average electron population of -0.16 to -0.17 e. All this reasoning supports the experimental assignment that the molecules **1** and **2** are physisorbed, not chemisorbed, to the substrate. (Supplementary Information Pg S19-S20)

3./ most of the discussion is related to the relationship between oxidation state and structural conformation. In this sense, it would be more interesting to move Fig. 4 to the supplementary (does not provide any information and STM-DFT calculations are always tricky) and replace it by fig. S17. When the structure of the different molecules can be seen.

3.1/ with respect to this figure S17, I understand that calculations were performed using three substrate layers. Is this thickness enough to ensure a good structure determination?

RESPONSE: The geometric models for molecule-surface interaction have been chosen to describe a range of interactions. Among these, the most important are: (i) localized chemical bonds (ii) molecular state-band interaction (delocalized) (iii) physical interactions (van der Waals, electrostatic etc). The first category can be described by using small clusters / single layers (even single atoms may capture an important part of chemistry occurring at surfaces). In order to include the surface band structure needed to estimate the physical interactions a 'slab' with a suitable thickness and lateral extension needs to be chosen. Cu(111) like any noble metal hosts a Shockley surface state which coincides with a gap in the bulk band structure. Therefore, a rather thin slab (three layers) is expected to be sufficient to adequately describe the interactions mentioned above.

Action taken: Figure S17 has been moved to replace Fig. 4 in the main manuscript as suggested by the reviewer. A discussion of the new Figure 4 (which has been updated) has been added to the main text (Pg 16). Figure 4 of the original submission (DFT STM simulation figure) showed the same information as the current Supplementary Fig. S15 and has been deleted.

Text added to main manuscript after placement of Fig. 17 (now Figure 4) into the main manuscript.

(Includes response to reviewer 3's questions)

The DFT structures of the three closely related acene species (Fig. 4) reveal the complexity of molecular adsorption based on the oxidation state: **2** is primarily adsorbed via the terminal phenyl groups interacting in a London-type (π -metal) interaction with the metal substrate. In contrast, in **2-ox** the molecular backbone is twisted to accommodate interactions between the substrate and the pyridinic N electron pairs on the acene. Note that acenes are known to be able to accommodate significant twisting of the backbone. Thus, the modification of the interaction of the Nitrogens is plausibly weak and essentially undetectable by XPS in spite of a slight broadening (see also Fig 2g,h,i and Supplementary Fig. S6g,h,i). Also note that this interaction does not stop the lateral movement of molecules on the surface, which is required to form the self-assembled chain structures. This configuration appears to be the prerequisite for the observed cooperative conformational adaptation and aggregation into the chain form (Fig. 2). This process is favored since intermolecular C-H...N interactions within the chains allow the relaxation of twisting of the acene backbone. Following cyclodehydrogenation, the adsorption is close to planar with the feature that the center of the backbone more closely approaches the surface in agreement with an earlier report on pentacene.

3./ It is surprising that 2-ox is a bit tilted, whereas 2 is not. Is any chemical bond between the N atoms and the surface (which are the involved distances from N to the surface) ? Shall this interaction be therefore reflected in the XPS spectrum?

RESPONSE: The main manuscript text has been updated in response to these questions on the adsorption of 2-ox in DFT. The twisted molecule is held on the substrate via the interaction of the C-N-free electron pairs. In spite of the considerable symmetry breaking introduced by this twist (See Fig. 4b), this associative bonding is expected to be rather weak and does not affect the XPS line shifts. This is also confirmed in the Voronoi charge analysis now provided in Supplementary Fig. S16 and the accompanying text. It is however to be noted that there are subtle changes between the line shapes of the 2-ox and the 2-ox-2 which may indicate a weak broadening of the N XPS for the twisted 2-ox in agreement with the sharper N1s line shape of the adsorbed 2-ox₂ as it appears consistently planar in STM and DFT.

3./ An important point related with my query 2.2, is about the charge transfer from the surface, if any. Is there any specific and different charge transfer associated to each oxidation state? In literature, some molecules have been found to curve when accepting charge on the surface. Is this the case?

RESPONSE: The proposal of the reviewer to investigate this effect further for the present case is a very good one. The Voronoi charge analysis has not given significant evidence in favour of a charge transfer / strong interaction. Notably 1 and 2 and their oxidized forms – like pentacene as an analogue -- are rich in pi-electrons and can undergo polarization which may lead to a ‘London-type’ of interaction which can be stronger than covalent bonds even in absence of hybrid electronic states. This case has been discussed in K. Mueller et al. [dx.doi.org/10.1021/jp308058u](https://doi.org/10.1021/jp308058u) | J. Phys. Chem. C 2012, 116, 23465–23471 and the refs

cited therein and could be investigated in further details. Presently, our data does not allow us to discriminate polarization and charge transfer effects in sufficient detail for the current system.

Action taken: Text has been added to the manuscript to clarify this issue. (Pg 15-16.)

4./ I cannot see the relation between the photo redox studies and the synthesis on the surface. It is a completely different characterization. The important point is that these wet routes does not produce 1-ox2 and 2-ox2 species. In this sense, the photoredox catalysis paragraph can be removed from the main text.

RESPONSE: Although the photoredox activity is an important aspect of the reduced compounds, it is also true that 1-ox2 and 2-ox2 species are only accessible so far by on surface routes making that a more important point as the reviewer suggest.

Action taken: Text describing the photoredox activity has been shifted to the Supplementary Information. Other references to the photoredox aspects have been removed or reduced.

5./ Transition from isolated molecules on the surface to ordered aligned structures is very interesting. It is not a polymerization process (line 353), and I wonder which is the role of diffusion or surface-molecule interaction for this structural transformation. Why 2 and 2-ox2 does form individual structures whereas 2-ox1 self-assemble in aligned structures, is it related with the molecular tilt? There is some pi-pi interaction between adjacent molecules?

RESPONSE: As mentioned above, it is possible that species 2 and 2-ox2 are physisorbed through interactions involving aromatic-metal substrate interactions and are consequently immobile. Molecular mobility for 2-ox is increased sufficiently such that it can aggregate into the observed chains. Neither DFT analysis nor XPS support directional chemical bonding in the form of N-surface interactions. The close-shell character of all discussed compounds is the basis for a physisorption dominated inter-molecular and molecular-substrate interactions. Inter-molecular condensation occurs only for 2-ox as it is more flexible allowing for two neighboring molecules to optimize their inter-molecular interaction by cooperative conformational adaptation maximizing the C-H...N interaction. This appears not to be the case for 2, neither for 2-ox-2 which both predominantly interact with the substrate via their end-groups locking down into the substrate. This view is supported both by experimental STM imaging as well as by DFT calculations on the individual molecules.

Action taken: Brief text discussing this has been added. Pg 15-16.

REVIEWERS' COMMENTS:

Reviewer #2 (Remarks to the Author):

The authors have addressed my comments carefully and correctly. It can be accepted now

Reviewer #3 (Remarks to the Author):

The authors have satisfactory addressed all my demands, and I consider that the manuscript can be accepted for publication.